# An NLP Benchmark Dataset for Evaluating the Completeness of ESG Reports

## Abstract

Environmental, Social, and Governance (ESG) reports serve as a platform for companies to publicly disclose their economic, environmental, and social impacts, as well as their contributions to sustainable development goals. The completeness of ESG reports is considered a crucial criterion for judging their quality and credibility, yet it is often overlooked in existing literature. This paper aims to comprehensively assess the completeness of ESG reports by evaluating their topic coverage and text quality. To achieve this goal, we collect 14,468 ESG reports from Chinese-listed companies. We then segment these reports into sentences and label over 8,000 of them with both topic and text quality tags. Finally, we propose two classification tasks based on the ESG sentences: topic classification and quality classification, to evaluate the ESG completeness. To train the classifiers, we fine-tuned several large language models (LLMs) on this dataset for the two classification tasks. Our findings suggest that the dataset has the potential to fill the gap in academia regarding methods for measuring ESG completeness.

## 1 Introduction

With the increasing awareness of sustainable development in society, how companies balance economic benefits with environmental benefits and social benefits has garnered close public attention. In this context, corporate environmental, social, and governance (ESG) performance has become a rapidly evolving focus [21, 18, 37]. Currently, ESG reports are a crucial means for companies to disclose their ESG performance, providing essential information for investors and stakeholders seeking insights into a company's commitment to these areas [31, 20].

Concerns have been raised regarding the ability of ESG reports to accurately reflect a company's contributions towards sustainable development [25, 27, 32]. Skeptics argue that ESG reports may act as a form of decoupling—a symbolic practice that is disconnected from actual performance, such as selective disclosure [25, 5, 36]. Selective disclosure, as shown in Figure 1, refers to the practice where companies disproportionately highlight favorable or relatively benign performance indicators to obscure their overall less impressive performance, thereby seeking to gain or maintain legitimacy [25]. Authors in [32] found that decoupling is prevalent in sustainability reports, with 69% of negative events being selectively reported. Numerous non-profit organizations and NGOs, such as the Global Reporting Initiative (GRI), the Sustainability Accounting Standards Board Foundation (SASB), and Bloomberg, introduced specific ESG indicator systems to mitigate this issue. These systems aim to clarify the essential ESG topics that companies should disclose, ensuring the completeness of ESG reports [30].

Completeness is a crucial criterion for assessing the quality of ESG reports [33]. It requires companies to comprehensively disclose significant economic, environmental, and social impacts related to their

Submitted to the 38th Conference on Neural Information Processing Systems (NeurIPS 2024) Track on Datasets and Benchmarks. Do not distribute.

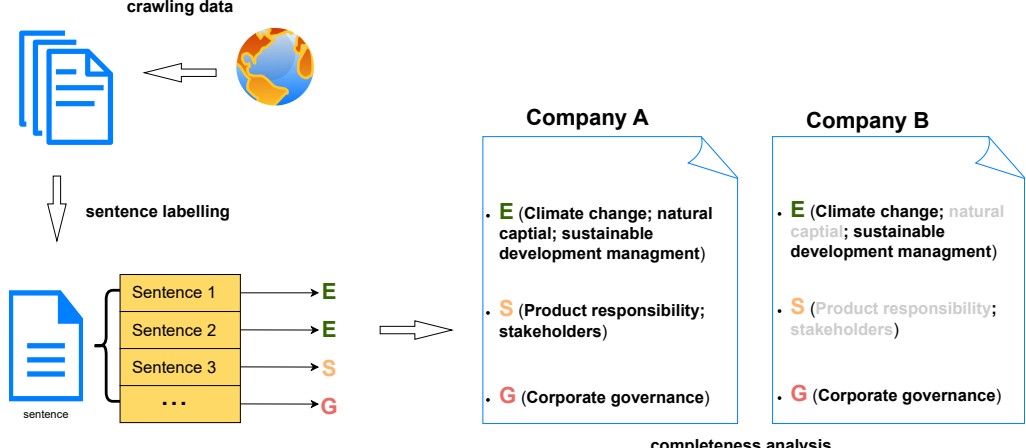

Figure 1: Dataset Collection, Labeling, and ESG Report Completeness Analysis. We collected ESG reports of publicly listed companies from the Internet. Each sentence in these reports is labeled with 36 categories of ESG tags. Using these ESG tags, we performed a quantitative analysis of the completeness of the reports. The completeness panel illustrates two examples: Company A, which exhibits a comprehensive ESG report, and Company B, which selectively discloses information by omitting categories such as "natural capital", "product responsibility", and "stakeholder".

operations [26]. Scholars emphasized that ESG reports are credible only when they meet completeness requirements [33, 26]. However, the academic community lacks scientific methods for evaluating ESG completeness. Additionally, international regulatory rules, such as the European Union's Corporate Sustainability Reporting Directive (CSRD), mandate that complete ESG reports should include both quantitative and qualitative information [2]. Furthermore, many countries and international stock exchanges encourage issuers to prioritize quantitative information in their ESG reports [6, 24]. Therefore, when studying the completeness of ESG reports, attention should be paid not only to the topics covered but also to the quality of their content.

Due to the diverse categories and extensive content covered by ESG topics [10], large-scale monitoring and identification of the completeness of ESG reports are extremely challenging, requiring domain experts to analyze company documents. This necessitates the construction of open, high-quality datasets suitable for training and evaluating models in such contexts, which can be alleviated through the rapid text classification processes facilitated by natural language processing (NLP).

However, it is important to note that existing datasets do not support research on the completeness of ESG reports. Although existing works provided datasets in the field of sustainability, they only focus on part of the ESG topics, such as climate and environmental areas [38, 27]. Additionally, many studies utilize unsupervised models like Latent Dirichlet Allocation (LDA) [3] to learn topic structures, relying on word co-occurrence trends [14, 16]. However, LDA is an unsupervised model with significant uncertainty in the number and criteria of clusters, meaning the topics generated and interpreted by one researcher may not completely align with those of another [4]. Hence, a fine-grained and labeled ESG dataset is essential for evaluating ESG completeness.

To address this need, we introduce a comprehensive dataset representing corporate ESG engagement, compiled from a wide range of company-related documents as illustrated in Figure 1. This dataset facilitates the detection of ESG report completeness, the generation and optimization of ESG reports, the evaluation of stakeholder assessments of corporate sustainability strategies, and the support of ESG fund investment decision-making systems. Initially, we evaluate the completeness of ESG reports based on topic coverage and disclosure quality, establishing an ESG tree and a two-tier classification system for ESG text quality. Utilizing this framework, we collected all ESG reports from Chinese-listed companies spanning the period, sourced from the official website of the Chinese stock exchange. We manually annotated 8,467 text sentences, each assigned two types of labels:

a topic label and a quality label. The topic labels are categorized into 36 classes according to the ESG tree, encompassing various aspects such as climate change, employee health and safety, and community engagement. The quality labels are divided into two categories: quantitative description and qualitative description.

The contributions of this work can be summarized as follows:

- Utilizing a scientific approach to evaluate ESG completeness in terms of both topic coverage and text quality.

- Introduction of a novel, fine-grained ESG dataset for evaluating the completeness of ESG reports and detailed manual annotation of text sentences with both topic and quality labels. This dataset is expected to stimulate research in natural language processing, sustainability, and ESG, guiding more accurate detection of ESG report completeness and evaluating corporate contributions to sustainability.

- We evaluate the performance of pre-trained language models and large language models on this task. Although we obtained promising results, such as an accuracy of approximately 85.66% in evidence page detection, there remains substantial room for improvement in evaluation performance. The code and dataset are available at `https://github.com/LCYgogogo/ESG-dataset`.

## 2 Background

**Selective disclosure issues in ESG reports**  ESG reports serve as instruments for measuring, disclosing, and communicating information related to corporate social responsibility and sustainability objectives [1, 13, 15]. These reports encompass a range of topics, including specific initiatives, significant risks, and policy goals undertaken by companies across ESG dimensions [1, 13, 15]. However, due to the lack of mandatory ESG reporting frameworks and strong government regulations worldwide, there are significant differences in the quantity, reporting formats, and content of ESG reports disclosed by companies [12]. Additionally, managers often have opportunistic motives for selectively disclosing information [25]. Consequently, the completeness of ESG reports has been questioned [28]. While many companies report substantial ESG information on various topics, the information is often one-sided, lacking disclosure on key ESG issues [29]. Some companies focus excessively on key dimensions related to their business operations while neglecting other CSR topics [29].

**NLP Research Related to the ESG completeness**  Existing works examined the completeness of ESG reports by analyzing the coverage of ESG topics [26, 16, 22]. In [16] and [22], researchers utilize unsupervised models, such as LDA, to learn the topic structure and cluster ESG texts, subsequently analyzing the content and trends of various topics. For instance, [16] revealed that ESG information disclosed by publicly listed companies in the UK and Europe primarily focuses on employee safety, employee training support, carbon emissions, human rights, efficient electricity, and healthcare products. However, as an unsupervised model, LDA presents significant uncertainty in generating and interpreting text topics [4]. Consequently, these methods are ineffective in evaluating the completeness of ESG report topics or identifying selective disclosure behaviors by listed companies.

## 3 Dataset

Our dataset assesses ESG report completeness from two perspectives: topic coverage and text quality. It offers valuable insights for various research applications. In the field of NLP, it encourages the application of NLP technologies in sustainable development. In the domains of sustainability and finance, models trained on our dataset can evaluate the completeness and credibility of a company's ESG reports, thereby informing investment decisions for ESG funds.

## 3.1 Dataset Construction

As shown in Figure 1, we evaluate the ESG completeness for each ESG report (document) $\mathbf{X}^i$ from two perspectives: topics and quality. To achieve this, we segment $\mathbf{X}^i$ into sentences and labeled each sentence with both topic and quality tags. Suppose report $\mathbf{X}^i$ contains $n_i$ sentences. Thus, $\mathbf{X}^i := \left\{ \mathbf{x}_j^i \right\}_{j=1}^{n_i}$, where $\mathbf{x}_j^i$ is the $j$-th sentence of report $i$, with $j = 1, 2, \ldots, n_i$. For each sentence $\mathbf{x}$, we assign two kinds of labels: one is the topic label $\mathbf{y}$, and the other is the quality label, denoted by $\mathbf{z}$.

We believe that both $\mathbf{y}$ and $\mathbf{z}$ contribute to the completeness of $\mathbf{X}^i$. The topic label $\mathbf{y}$ is a 36-dimensional one-hot vector corresponding to the leaf nodes of the ESG tree shown in Figure 2. Details regarding the 36 topic labels will be discussed in the next section. The quality label $\mathbf{z}$ is a 2-dimensional one-hot vector representing "Quantitative description" and "Qualitative description".

We collect ESG reports $\mathbf{X}^i$ released by Chinese-listed companies from the official website of the China Stock Exchange, resulting in a total of 14,468 documents. Following the definitions of the labels $\mathbf{y}$ and $\mathbf{z}$, we engage three Ph.D. researchers specializing in the ESG domain to annotate training sets for the 36 topic labels $\mathbf{y}$ and the 2 quality labels $\mathbf{z}$. This process results in 8,467 manually labeled text sentences. We exclude 483 irrelevant ones, such as tables of contents and acknowledgments, which are unrelated to the ESG topic content. Subsequently, we assign two labels to the remaining 7984 sentences: the topic and quality labels. Consequently, we obtain the dataset $\mathcal{D} := \{\mathbf{x}_j, \mathbf{y}_j, \mathbf{z}_j\}_{j=1}^{8467}$, as illustrated in Table 1. The average length of these sentences is 80 Chinese characters. By segmenting the unlabeled ESG reports, we obtained over 3.2 million sentences, forming the out-of-distribution sample set.

| | Train | Test | ESG Class | Quality Class | Average Len | Out-of-distribution samples |
|---|---|---|---|---|---|---|
| sentences | 6,773 | 1,694 | 36 | 2 | 80.54 | 3,216,968 |

Table 1: Dataset description.

## 3.2 Two types of ESG label and ESG completeness

**ESG topic label** We use the ESG tree, as shown in Figure 2, to define the completeness with the topic classification. The topic labels $\mathbf{y}$ correspond to the leaf node of ESG tree. We construct the ESG tree according to the standards of internationally recognized third-party organizations, including GRI, SASB, the Carbon Disclosure Project (CDP), Morgan Stanley Capital International (MSCI), Bloomberg, the China Securities Index (CSI), and SynTao Green Finance (SGF).

Figure 2 illustrates the four-layer ESG tree we constructed, a hierarchical framework that dissects corporate sustainability into Environmental, Social, and Governance dimensions, each further divided into related sub-topics. For example, the second-level indicator "*Environment*" includes three third-level indicators: *climate change*, *natural capital*, and *sustainable development management*. Furthermore, for *climate change*, the leaf nodes are *carbon emissions* and *response to climate change*.

The ESG tree incorporates the disclosure requirements mandated by Chinese regulatory authorities for listed companies' ESG reports. For instance, the China Securities Regulatory Commission encourages listed companies to disclose their contributions to rural revitalization in China. To align with this requirement, we include "Rural Assistance" as a third-level topic. For detailed sources of the ESG tree labels, please refer to Appendix 1.

**Text quality label** We define ESG text quality through two types of labels. Based on our literature analysis, international authoritative ESG rating agencies, national securities regulatory authorities, and international stock exchanges increasingly emphasize that ESG reports should include crucial quantitative data in addition to qualitative descriptions [17]. Furthermore, there is growing encouragement for disclosing quantitative information [35]. Therefore, we examine the quality of ESG text as a crucial component in assessing the completeness of ESG reports. We categorize ESG text quality into two classes: (1) "Quantitative Text", which reflects quantitative information about the

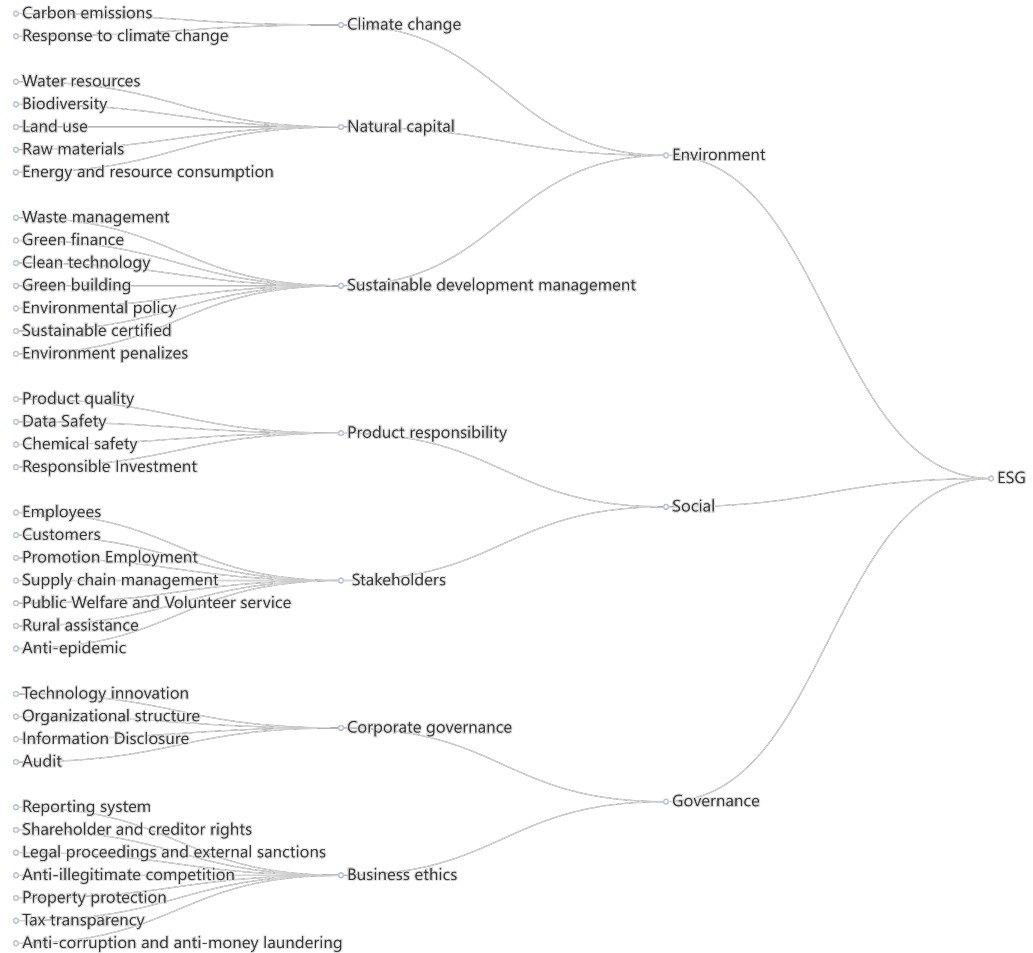

Figure 2: This ESG tree aids in the meticulous and systematic analysis of ESG topics. The topic hierarchical division of the ESG tree is derived from the standards of multiple ESG rating organizations (see subsection 3.2). Its 36 leaf nodes correspond to our 36 categories for sentence topic classification tasks.

ESG aspects of the company, and (2) "Qualitative Text", which reflects qualitative information about the ESG aspects of the company.

**ESG completeness evaluation**    The completeness of ESG reports can be evaluated using a weighted topic distribution derived from the results of topic classification and text quality classification, as illustrated in Figure 4. This approach involves projecting each sentence of an ESG document onto a corresponding topic label and then weighting these labels based on text quality. For instance, we assign scores of 2 to "Quantitative" sentences and 1 to "Qualitative" sentences. Thus, for a specific topic in an ESG report that contains one "Quantitative" sentence and one "Qualitative" sentence, the topic frequency would be calculated as 2 + 1 = 3, rather than simply 2.

## 4    Experiment

In this section, we evaluate our method for assessing ESG completeness on the constructed dataset. We employ several large language models and fine-tune them on this dataset to classify both topic and text quality.

### 4.1 Setups

We adjust the learning rate according to the complexity of the task. Specifically, the learning rate is 2e-5 for the quality classification task and 1e-4 for the text topic classification task. We use Adam [19] with a weight decay rate of 0.1, and stop training if the test loss does not decrease for 3 consecutive epochs. The batch size is 16. The fine-tuned models train 100 epochs, with a maximum sequence length of 512, the ablation study on the PEFT methods is detailed in Appendix 4. See Appendix 2 for more details on the pre-trained models. We train all models on an A100 GPU.

### 4.2 Baseline

- BERT [11]: A milestone in the field of NLP, which learns language representation through pre-trained and fine-tuned and utilizes two pre-trained tasks, Masked Language Model (MLM) and Next Sentence Prediction (NSP), it has significantly advanced the performance across a broad spectrum of NLP tasks.

- RoBERTa [23]: A variant of BERT that optimizes the original pre-trained methods, including scaling and complicating the training data, as well as improving the dynamic masking mechanism.

- LERT [7]: A novel pre-trained model that enhances linguistic feature learning by incorporating three types of linguistic features into the traditional masked language model task.

- PERT [8]: A pre-trained model based on an out-of-order language, introduces an auto-encoding mechanism with a Permuted Language Model (PerLM) objective, combining whole word and N-gram masking techniques to enhance performance.

- LLaMA2 [34, 9]: It introduces the Grouped Query Attention (GQA) mechanism during the supervised fine-tuning (SFT) stage, significantly enhancing inference efficiency and scalability in large models. Additionally, in the reinforcement learning phase, LLaMA2 employs the Grouped Attention (GAtt) mechanism to effectively address the issue of context forgetting.

### 4.3 ESG topic classification results

We present the comprehensive performance of ESG topics on the testing dataset, as detailed in Table 2, which compares the models without fine-tuning against those with fine-tuning. Row 2 of Table 2 presents the performance for the baseline models without fine-tuning. The results indicate that all models perform poorly on the ESG topic classification. For instance, the PERT (base) model has the lowest accuracy at 0.53%, while BERT and RoBERTa (large) achieve 0.65% and 0.54%, respectively. Row 3 displays the metrics for the models after fine-tuning. Fine-tuning significantly enhances performance across the board. For instance, the BERT model's accuracy increases from 0.65% to 81.28%, and the LERT (large) model improves from 1.59% to 84.18%. These improvements underscore the importance of fine-tuning in adapting the models to the specific domain of ESG topics.

Among the fine-tuned models, LLaMA2 exhibits the highest accuracy at 85.66%. The performance suggests that fine-tuning is particularly well-suited for the ESG topic classification task. Additionally, RoBERTa (large) and LERT (large) also show strong performance with accuracies of 84.36% and 84.18%, respectively.

### 4.4 Prompt results

We investigate the impact of different prompt designs on the performance of the LLaMA2 model in two tasks, topic classification and quality classification. As indicated in Table 3, prompt design significantly affects model performance. Notably, in the topic classification task, the prompt 3 design, substantially improved the model's accuracy. The accuracy for topic classification with prompt 1 is merely 77.27%. Still, with prompt 3, the accuracy rose to 85.66%, which is significantly higher than other prompt designs. It suggests that carefully crafted prompts can greatly enhance the model's

| Fine-tuned | metrics | BERT | LERT (base) | LERT (large) | PERT (base) | PERT (large) | RoBERTa (base) | RoBERTa (large) | LLaMA2 |
|---|---|---|---|---|---|---|---|---|---|
| ✗ | Precision | 0.54% | 0.43% | 0.50% | 0.01% | 0.02% | 0.09% | 0.03% | 1.11% |
| | Recall | 2.28% | 1.82% | 2.86% | 2.70% | 2.70% | 2.89% | 2.38% | 2.75% |
| | F1 | 0.30% | 0.42% | 0.51% | 0.03% | 0.03% | 0.16% | 0.06% | 0.68% |
| | Accuracy | 0.65% | 1.65% | 1.59% | 0.53% | 0.59% | 0.71% | 0.54% | 1.77% |
| ✓ | Precision | 79.25% | 79.41% | 83.21% | 64.62% | 73.91% | 81.36% | 84.99% | 85.25% |
| | Recall | 74.09% | 72.87% | 75.46% | 62.22% | 69.04% | 77.39% | 78.69% | 80.08% |
| | F1 | 75.08% | 74.09% | 78.08% | 62.00% | 69.20% | 78.74% | 80.87% | 81.54% |
| | Accuracy | 81.28% | 82.47% | 84.18% | 77.92% | 79.93% | 83.18% | *84.36%* | **85.66%** |

Table 2: Performance of fine-tuned large language models on ESG topic classification. We evaluate the performance impact of fine-tuning on different language models. Fine-tuning requires additional training on the ESG dataset to improve performance. The best results are highlighted in **boldface** and the second in *italic font*.

| | Topic classification | | | Quality classification | | |
|---|---|---|---|---|---|---|
| | prompt 1 | prompt 2 | prompt 3 | prompt 4 | prompt 5 | prompt 6 |
| Precision | 78.69% | 81.44% | 85.25% | 69.50% | 79.07% | 89.52% |
| Recall | 68.52% | 75.54% | 80.08% | 76.58% | 72.56% | 62.32% |
| F1 | 70.70% | 77.53% | 81.54% | 68.37% | 74.41% | 61.08% |
| Accuracy | 77.27% | 82.76% | **85.66%** | 80.11% | **88.72%** | 86.66% |

Table 3: Performance of prompt designs for LLaMA2 on topic classification and quality classification. To evaluate the effectiveness of different prompt designs, we devise three distinct prompts for each task, as detailed in Appendix 5.

performance on complex tasks. For the quality classification task, the influence of prompt design is also significant, prompt 5 achieves a significant enhancement, reaching 88.72%.

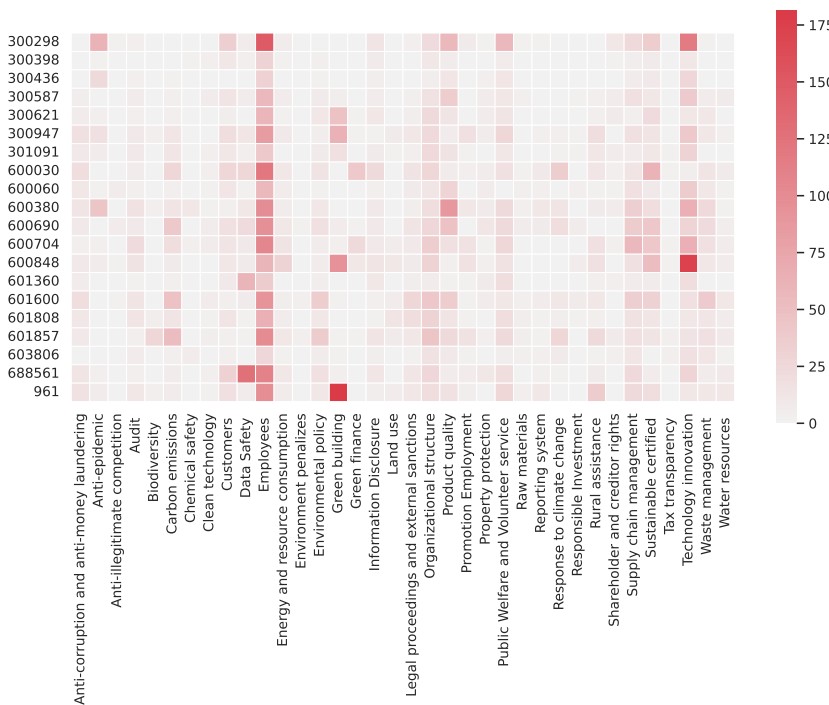

Figure 3: The heatmap visualizes the sentence count distribution across various ESG topic in the annual reports of 20 companies for the year 2022, post-elimination of irrelevant content. Identified by their stock codes on the vertical axis and arrayed the ESG topic on the horizontal axis.

| ESG sentences | Quality w/o fine-tuning | Quality w/ fine-tuning |
|---|---|---|
| The scale of international oil and gas cooperation continues to expand, with further improvements in operational quality. | quantitative text | qualitative text |
| Conducted risk compliance training 45 times, carried out 270 audit projects, undertook first-issue learning sessions 1,243 times, and responded to 97 questions on the Shanghai Exchange E-Interaction Platform. | quantitative text | quantitative text |
| Incorporate environmental protection and resource conservation into product design, selection of raw and auxiliary materials, processing, warehousing, and production and transportation. Additionally, waste materials resulting from the use of raw and auxiliary materials are recycled and reused. | irrelevant text | qualitative text |
| This report is compiled in Chinese, and the English version is provided for reference only. In case of any discrepancy in meaning between the English and Chinese versions, the Chinese version shall prevail. | quantitative text | irrelevant text |
| We carried out professional knowledge explanations on the two topics of font infringement and trademark use for employees, covering the serious consequences of font infringement, methods for determining the commercial use of fonts, and the legal use and transfer of trademarks. | quantitative text | qualitative text |

Table 4: ESG Text quality predictions on unlabelled annual reports. Columns 2 and 3 are the results of the baseline and proposed method respectively.

| ESG sentences | Topic w/o fine-tuning | Topic w/ fine-tuning |
|---|---|---|
| The information and data disclosed in this report are derived from the company's statistical reports and official documents, and have been reviewed by the relevant departments. | Land use | Information Disclosure and Communication with investors |
| Since publishing its inaugural ESG report in 2022, Zhongnan Construction has garnered 49 awards related to ESG, with its ESG practices also receiving ongoing close scrutiny from capital market rating agencies. | Green building | Sustainable certified |
| A total of 51 director participations were recorded in anti-corruption training, while employees accumulated over 36,000 courses hours in online anti-corruption training. | Organizational structure and operation | Anti-corruption and anti-money laundering |
| Conducted three rounds and four iterations of house quality inspections involving 122 components and 1,300 detailed inspections items, and continued to carry out pre-improvement project quality control actions such as Operation Eagle Eye on process and delivery assessments. | Organizational structure and operation | Product quality |
| During the reporting period, the Zhongnan audit system has fully covered the entire business process from front-end investment and land acquisition, mid-end project operation to back-end sales management, isolating the company from potential business risks and management risks. | Information Disclosure and Communication with investors | Audit |
| In 2022, the company continued to increase its R&D investment, with total research and development expenses amounting to RMB 259.8141 million, representing 9.24% of the revenue from operations. | Information Disclosure and Communication with investors | Technology innovation |

Table 5: ESG topic predictions on unlabelled annual reports. Columns 2 and 3 are the results of the baseline and proposed method respectively.

## 4.5 Predictions on future data

**Classification performance** To evaluate the classification capabilities of the fine-tuned large language model, we extract text from 20 unlabelled ESG reports from 2022 for sentence segmentation and compare the model's predictive performance on sentence labeling before and after fine-tuning.

Table 4 are the results of classifying sentences into "quantitative text" or "irrelevant text" before fine-tuning, while after fine-tuning, it shows a propensity to identify "qualitative text", indicating a deeper comprehension of text quality stratification. Based on the results of the quality classification task, we delete irrelevant texts and then classify qualitative and quantitative texts on ESG topics. The topic classification task, as shown in Table 5, we find that the model initially performed poorly, aligning with previous accuracy results in Table 2, tending to assign more generic labels such as "Information Disclosure and Communication with Investors" or "Organizational structure and operation". The model demonstrated a significant improvement in predictive performance with fine-tuning, capable of discerning more nuanced and specific labels like "Information Disclosure and Communication with investors", "Sustainable certified", "Anti-corruption and anti-money laundering", "Audit", and "Technology innovation". This improvement suggests that the LLaMA2 model, after fine-tuning, has notably advanced in the accuracy and granularity of predictive labeling, more precisely capturing the specific meanings and quality characteristics of sentences.

**Visualizations of completeness of ESG reports** The heatmap as shown in Figure 3 displays the distribution of sentence counts across 36 topics for 20 companies. Each row represents a company, identified by its code, and each column corresponds to the topic. The intensity of the color indicates the number of sentences in that category for the corresponding stock, with darker colors indicating higher quantities. Notably, certain topics show a high concentration of sentences for "Employees", which is consistent with the ESG report.

Figure 4 presents a comparison of sentence frequencies across the 36 topics. The bar chart displays the number of sentences for each category, while the line chart shows the cumulative distribution of sentence counts. In the bar chart, each color represents a stock with the number of sentences in each category depicted by bars of corresponding colors, and the line chart uses lines in matching colors to represent the cumulative distribution of sentence counts for each stock. Additionally, we visualize

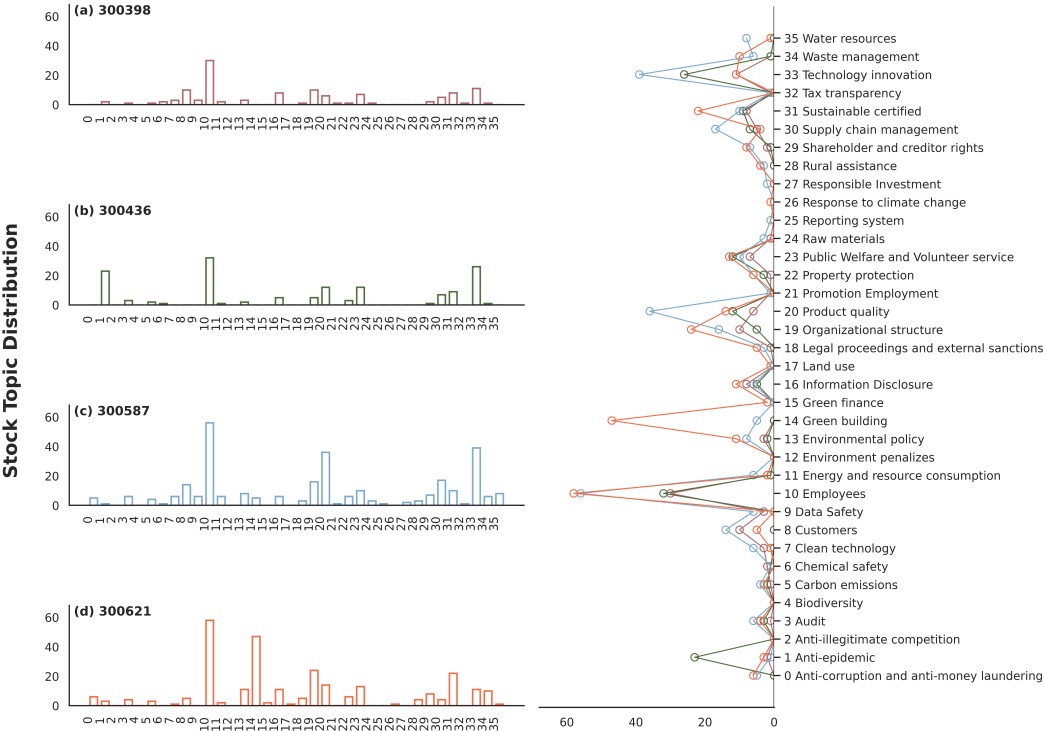

Figure 4: The topic distribution for stocks (listed companies): 300398, 300436, 300587, 300621. The left side shows bar charts detailing the ESG topic distribution. On the right, the line chart compares sentence frequencies, revealing the diverse focus each company has on the ESG topic.

the quality label distribution and topic classification using a sunburst chart, as shown in Appendix 6.1 and 6.2, respectively.

## 5 Conclusion and limitation

**Conclusion** Research on utilizing NLP to assess the completeness of ESG reports is still in its early stages. We present a novel NLP dataset specifically designed to evaluate ESG completeness. To facilitate this, we establish topic and quality labels using high-dimensional vectors for classification purposes, and annotate the dataset accordingly. The fine-tuned LLMs exhibit higher precision and robust applicability in evaluating the completeness of ESG reports. We anticipate that our dataset will stimulate further research in both NLP and sustainable development.

**Limitation** We manually annotate the themes and narrative quality of ESG sentences. However, due to limited manpower, the number of annotations remains insufficient. Consequently, the accuracy of text theme classification did not exceed 90%, impacting the assessment of ESG completeness. Moving forward, we plan to increase the number of annotations and implement an active learning strategy to enhance their quality. This approach aims to collectively improve the accuracy of text classification and achieve a more precise assessment of ESG completeness.

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
