# 1 Source of ESG Topic

| Pillars | Themes | Key Issues | Source |
|---|---|---|---|
| Environment | Climate change | Carbon emissions | CSI |
| Environment | Climate change | Response to climate change | CDP |
| Environment | Natural capital | Water resources | CDP |
| Environment | Natural capital | Biodiversity | CDP |
| Environment | Natural capital | Land use | GRI |
| Environment | Natural capital | Raw materials | GRI |
| Environment | Natural capital | Energy and resource consumption | GRI |
| Environment | Sustainable development management | Waste management | GRI |
| Environment | Sustainable development management | Green finance | CSI |
| Environment | Sustainable development management | Clean technology | GRI |
| Environment | Sustainable development management | Green building | New indicator |
| Environment | Sustainable development management | Environmental policy | CSI |
| Environment | Sustainable development management | Sustainable certified | CSI |
| Environment | Sustainable development management | Environment penalizes | CSI |
| Social | Product responsibility | Product quality | CSI |
| Social | Product responsibility | Data Safety | MSCI |
| Social | Product responsibility | Chemical safety | CDP |
| Social | Product responsibility | Responsible Investment | GRI |
| Social | Stakeholders | Employees | CSI |
| Social | Stakeholders | Customers | GRI |
| Social | Stakeholders | Promotion Employment | GRI |
| Social | Stakeholders | Supply chain management | SASB |
| Social | Stakeholders | Public Welfare and Volunteer service | GRI |
| Social | Stakeholders | Rural assistance | New indicator |
| Social | Stakeholders | Anti-epidemic | New indicator |
| Governance | Corporate governance | Technology innovation | MSCI |
| Governance | Corporate governance | Organizational structure and operation | GRI |
| Governance | Corporate governance | Information Disclosure and Communication with investors | Bloomberg |
| Governance | Corporate governance | Audit | SASB |
| Governance | Business ethics | Reporting system | SASB |
| Governance | Business ethics | Shareholder and creditor rights | SASB |
| Governance | Business ethics | Legal proceedings and external sanctions | MSCI |
| Governance | Business ethics | Anti-illegitimate competition | SGF |
| Governance | Business ethics | Property protection | MSCI |
| Governance | Business ethics | Tax transparency | SASB |
| Governance | Business ethics | Anti-corruption and anti-money laundering | CSI |

Table 1: ESG Topics and Key Issues.

We establish the ESG topics and key issues according to the standards of internationally recognized third-party organizations, including the Global Reporting Initiative (GRI)[1] [1], the Sustainability

---

[1]https://www.globalreporting.org/

Accounting Standards Board Foundation (SASB)[2] [5], the Carbon Disclosure Project (CDP)[3] [3], Morgan Stanley Capital International (MSCI)[4] [2], Bloomberg [5] [6], the China Securities Index (CSI)[6] [9], and SynTao Green Finance (SGF)[7] [10]. Additionally, in line with the regulatory requirements for ESG report disclosures by listed companies, and considering both the contextual background and the actual content disclosed by these companies, we include categories such as Green Building, Rural Assistance, and Rural Education as new indicators.

## 2 Model details

|  | LERT | PERT | BERT | RoBERTa (base) | RoBERTa (large) | LLaMA 2 |
|---|---|---|---|---|---|---|
| Hidden | 768 | 768 | 768 | 768 | 1024 | 4096 |
| Layer | 12 | 12 | 12 | 12 | 24 | 4 |
| Attention head | 12 | 12 | 12 | 12 | 16 | 32 |
| Trainable params | 0.1M | 0.1M | 0.1M | 0.1M | 0.2M | 0.4M |
| All params | 0.1B | 0.1B | 0.1B | 0.1B | 0.3B | 1.03B |
| Trainable params (%) | 0.10% | 0.10% | 0.07% | 0.09% | 0.07% | 0.04% |
| runtime(s) | 429.20 | 802.32 | 981.80 | 965.78 | 1107.13 | 5623.03 |

Table 2: Model detail.

All the foundational models utilized in this research are derived from the Chinese pre-trained models provided by HFL on the Hugging Face. The Chinese pre-training corpus encompasses a wide array of data resources, including but not limited to Chinese Wikipedia dump, encyclopedia, community question answering, and news articles. For detailed information and access to resources, please refer to `https://huggingface.co/hfl`.

In terms of trainable parameters in table Table 2, LLaMA2 boasts 2M trainable parameters, far exceeding the others, yet it has the lowest percentage of trainable parameters at just 0.03%.

## 3 Dataset details

In the quality classification task, figure Figure 1 demonstrates that the distribution of qualitative text significantly dominates the dataset, surpassing other texts by a considerable margin. The proportion of irrelevant text is relatively minimal, aligning closely with the structural composition of ESG reports. In the ESG reports, irrelevant text such as 'In case of discrepancies between the English and Chinese versions, the Chinese version shall prevail' and 'Please provide an overall evaluation of this report' constitutes only a minor share, while the majority of the content is predominantly qualitative.

Figure Figure 2 presents the dataset distribution for the topic classification task. To ensure accuracy and reliability in the ESG topic classification, we have labeled a minimum of 25 sentences for each topic.

---

[2]https://sasb.ifrs.org/

[3]https://www.cdp.net/en

[4]https://www.msci.com/

[5]https://www.bloomberg.com/

[6]https://www.csindex.com.cn/en#/

[7]https://www.syntaogf.com/

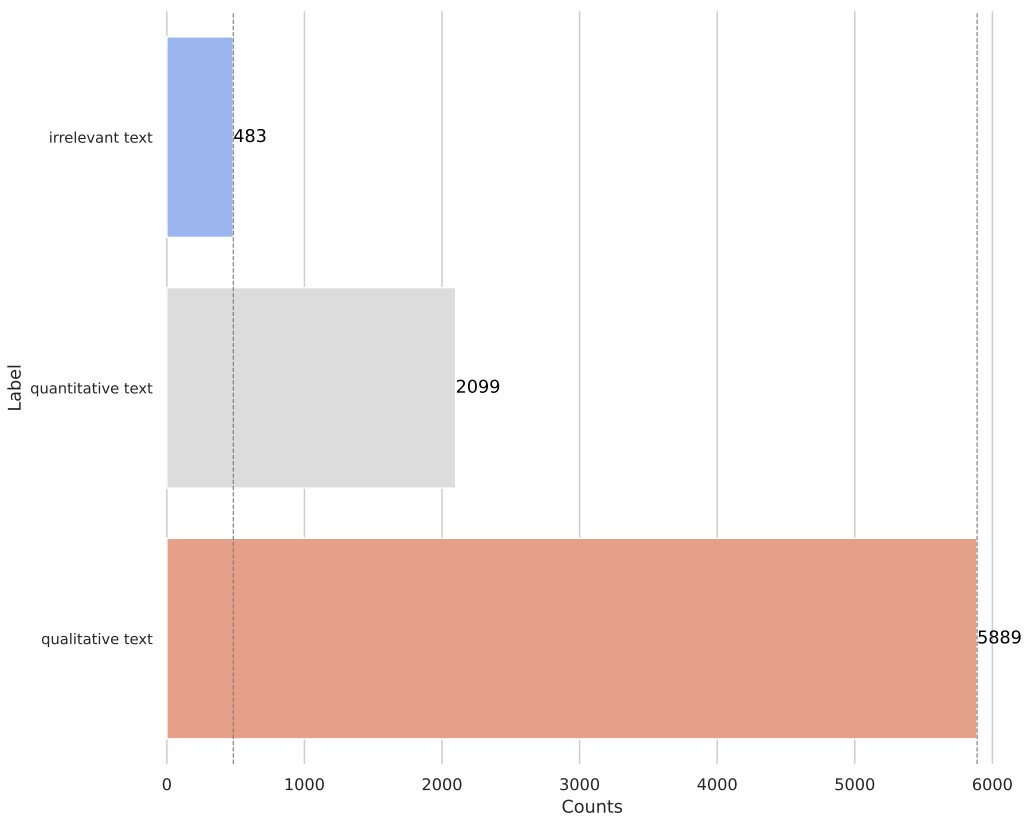

Figure 1: Distribution of qualitative text.

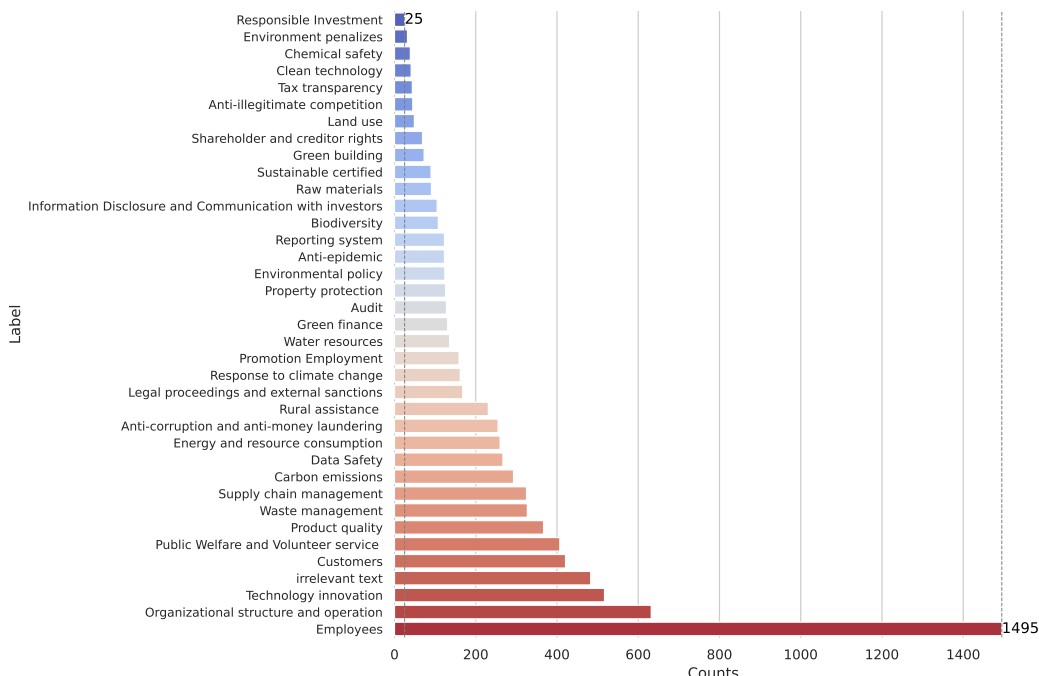

Figure 2: Distribution of topic text.

## 4 PEFT results

| | LoRA | P-tuning | Prompt-tuning(soft) | Prompt-tuning (hard) |
|---|---|---|---|---|
| Config | r = 4, $\alpha$ = 32 | tokens = 10 | tokens = 10 | prompt 3 |
| Precision | 83.56% | 81.42% | 78.37% | 85.25% |
| Recall | 77.72% | 75.22% | 71.22% | 80.08% |
| F1 | 80.01% | 77.14% | 73.66% | 81.54% |
| Accuracy | *84.89%* | 83.17% | 80.64% | **85.66%** |

Table 3: Different PEFT performance on LLaMA2.

We leverage Parameter-Efficient Fine-Tuning methods such as Prompt-Tuning [7], LoRA [4] and P-Tuning [8]. We conducted a comparative analysis of the performance of different PEFT methods based on LLaMA2. As shown in Table 3, we tested four fine-tuned configurations on the topic classification task. Employing various fine-tuning strategies on the LLaMA2 model has a marginal impact on performance. In this experiment, the Prompt-tuning (hard) method stood out, achieving the highest accuracy at 85.66%.

## 5 Prompt design

Prompt 1: Classify the ESG report text by topic.

Prompt 2: The ESG report comprises 3 first-level indicators, 7 second-level indicators indicators, and 36 third-level indicators. As a logical ESG report analyst, you classify the ESG text according to the third-level indicators.

Prompt 3: The ESG report comprises 3 first-level indicators, 7 second-level indicators, and 36 third-level indicators, organized in a tree structure:Environment: Climate Change: [Response to Climate Change, Carbon Emissions], Natural Capital: [Land Use, Biodiversity, Raw Materials, Energy and Resource Consumption, Water Resources], Sustainable Development Management: [Clean Technology, Sustainable Certification, Green Finance, Environmental Penalties, Green Building, Waste Management, Environmental Policy], Social: Product Responsibility: [Data Safety, Product Quality, Responsible Investment, Chemical Safety], Stakeholders: [Employees], Stakeholders: [Supply Chain Management, Employment Promotion, Rural Assistance, Public Welfare and Volunteer Service, Customers and Consumers, Anti-Epidemic Measures], Governance: Corporate Governance: [Audit, Organizational Structure and Operation, Technological Innovation, Information Disclosure and Investor Communication], Business Ethics: [Intellectual Property Protection, Legal Litigation and External Sanctions, Anti-Unfair Competition, Whistleblower System, Shareholder and Creditor Rights, Anti-Corruption and Anti-Money Laundering, Tax Transparency]. As a logical ESG report analyst, you classify the ESG text according to the third-level indicators.

Prompt 4: Classify the ESG report text by quality.

Prompt 5: As a logical analyst of ESG reports, you classify the quality of ESG texts into three categories: quantitative texts, which include numerically quantified ESG content; qualitative texts, referring to descriptive ESG narratives; and irrelevant texts, such as indexes, English sections, and other non-essential ESG content.

Prompt 6: The keywords related to ESG are:Environment: Climate Change: [Response to Climate Change, Carbon Emissions], Natural Capital: [Land Use, Biodiversity, Raw Materials, Energy and Resource Consumption, Water Resources], Sustainable Development Management: [Clean Technology, Sustainable Certification, Green Finance, Environmental Penalties, Green Building, Waste Management, Environmental Policy], Social: Product Responsibility: [Data Safety, Product Quality, Responsible Investment, Chemical Safety], Stakeholders: [Employees], Stakeholders: [Supply Chain Management, Employment Promotion, Rural Assistance, Public Welfare and Volunteer Service, Customers and Consumers, Anti-Epidemic Measures], Governance: Corporate Governance: [Audit, Organizational Structure and Operation, Technological Innovation, Information Disclosure and Investor Communication], Business Ethics: [Intellectual Property Protection, Legal Litigation and External Sanctions, Anti-Unfair Competition, Whistleblower System, Shareholder and Creditor Rights, Anti-Corruption and Anti-Money Laundering, Tax Transparency]. As a logical ESG report analyst, you classify the ESG text into two quality categories: (1) Quantitative text related to ESG that contains numerical quantification, (2) Qualitative text related to ESG that is descriptive.

Figure 3: The prompts used in this work.

The design of prompts is pivotal in training language models, particularly in specialized domains. We incorporate the ESG tree's structure into the prompts to enhance classification accuracy.

In the "Topic classification prompt design," we structured the prompts to guide the model through a hierarchical framework of ESG indicators. This framework includes 3 first-level indicators, 7 second-level indicators, and 36 third-level indicators. This detailed structure ensures that the model captures the comprehensive scope of ESG topics, facilitating accurate classification.

Similarly, the "Quality classification prompt design" focuses on distinguishing between irrelevant, qualitative, and quantitative texts. This differentiation is crucial for evaluating the completeness and integrity of ESG reports.

# 6 Visualizations of completeness of ESG reports

## 6.1 Visual quality classification

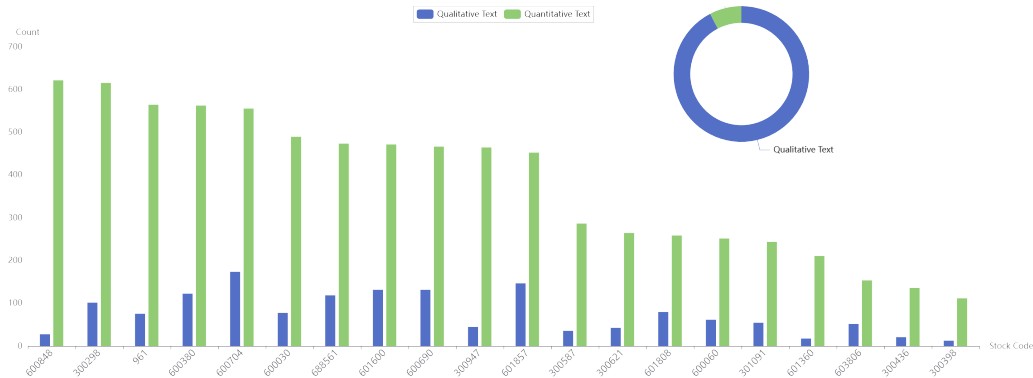

Figure 4: Quality classification prediction on 2022 ESG reports.

Figure 4 depicts the distribution of text quality in the 2022 annual reports for 20 stocks, as predicted by the fine-tuned LLaMA2. The figure provides a percentage breakdown of the text types across all reports, with qualitative text making up the majority, followed by quantitative text, and a minimal portion of irrelevant text. Additionally, the visualization is consistent with the description in Appendix B: dataset details.

## 6.2 Visual reports of stock 000961.

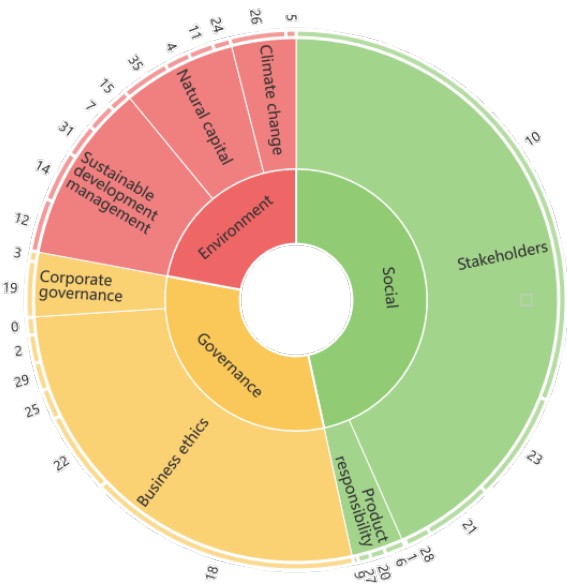

Figure 5: Sunburst on 2022 ESG reports of stock 000961.

In Figure 5, we present a topic classification through a sunburst, taking the annual report of stock code 000961 as a case to demonstrate the visualization of its Environmental, Social, and Governance

(ESG). The sunburst is structured in concentric circles, with the center circle representing the overall annual report. As the layers expand outward, they reveal a more detailed categorization, such as major categories including Environment, Social, and Governance, further divided into specific subcategories like Climate Change and Natural Capital. Different color segments indicate different categories, with the size of each segment reflecting the frequency of sentences within that category in the annual report. The company with the stock code 000961 pays greater attention to social and governance. The sunburst enables quick recognition of key areas and their comparative significance in the ESG disclosures.