# OpenReview forum: "An NLP Benchmark Dataset for Predicting the Completeness of ESG Reports"
_NeurIPS.cc/2024/Datasets_and_Benchmarks_Track — Submitted to NeurIPS 2024 Track Datasets and Benchmarks_

### Official Review · Reviewer_yB1f · 2024-07-08
**An NLP Benchmark Dataset for Evaluating the Completeness of ESG Reports**

**Rating:** 5
**Confidence:** 4
**Correctness:** Yes.
**Clarity:** Yes.

**Review:**

Please see above.

**Strengths:**

1. The paper introduces a large and comprehensive dataset. It contains 14468 ESG reports from Chinese-listed companies with over 8000 manually labeled sentences.
2. The research achieves an accuracy of approximately 85.66% in evidence page detection. This high accuracy indicates that the models can reliably identify and categorize relevant ESG information.

**Additional Feedback:**

NA.

**Documentation:**

Yes.

**Ethics:**

No.

**Limitations:**

Please see above.

**Opportunities For Improvement:**

1. The dataset used in the study is focused on ESG reports from Chinese-listed companies. The generalizability of the findings is limited. There can also be language and cultural bias from such data sources.
2. This research is built on the assumption that completeness is the most crucial criterion for assessing the quality and value of these reports. Most development efforts have been made for assessing completeness of the reports. However, I feel that this assumption is questionable. Accuracy, transparency, impact, etc. are also important for the reports. If completeness alone does not significantly improve the usefulness of ESG reports for stakeholders, then the practical value of this dataset is limited.
3. I appreciate the author(s)’s efforts spent in manual annotation for labeling sentences with topic and quality tags. While this ensures high accuracy, it also introduces scalability issues and the potential for human error or bias in the labeling process.
4. There could be more fine-tuning results for newer open-source models.
5. I suggest that the author(s) provide more insights on the numerical results, e.g., why the models performed as they did, potential reasons for any misclassifications, or the implications of the numerical results.

**Relation To Prior Work:**

Yes.

**Summary And Contributions:**

This research collected a dataset of ESG reports from Chinese-listed companies and segmented them into sentences, with over 8000 sentences labeled with both topic and text quality tags. Two classification tasks were proposed: topic classification and quality classification, to evaluate the completeness of ESG reports. Then, LLMs were fine-tuned on the dataset for classification tasks, and showed promising results in detecting ESG report completeness.

---

> ### Author Rebuttal · Authors · 2024-08-17
>
> Response to # yB1f
> #R4.A1:
> Our dataset fills a critical gap in Chinese ESG research, supporting deeper analysis of ESG reports and aiding the development of related laws in China. It also helps detect selective disclosure practices through meticulous labeling. With 36 carefully curated ESG sub-categories, our innovative combination of textual themes and quality enhances future ESG evaluations, setting a new standard for data comprehensiveness and serving as a valuable resource for both researchers and policymakers.
> #R4.A2:
> According to the International Integrated Reporting Council (IIRC) and the European Union's Corporate Sustainability Reporting Directive (CSRD), as well as related research, integrity is the most important standard for evaluating the quality and value of these reports (Kennedy Nyahunzvi 2013; Veltri 2020; Breijer and Orij 2022). The integrity of ESG reports is crucial for ensuring the comprehensiveness and accuracy of information, preventing selective disclosure and whitewashing, maintaining market fairness and investor interests, promoting corporate sustainability, and complying with regulatory requirements and industry standards (Kennedy Nyahunzvi 2013; Melloni et al. 2017; Eng et al. 2021; Darnall et al. 2022). Based on this, we evaluate ESG reports from the perspectives of textual themes and textual quality, grounded in relevant international standards and existing research.
> Of course, the accuracy, transparency, and impact of reports, as you mentioned, are also important, but there is currently no effective way to quantify them, nor is there relevant research to draw upon.
> #R4.A3: We made the following efforts to ensure the consistency and quality of dataset labels:
> First, two experts in law and finance who specialize in ESG, CSR, and green finance research coordinated the labeling work and developed our standards for this study (see appendix). These experts have published several influential research papers in international journals such as the Journal of Financial Econometrics, Management Decision, and Pacific-Basin Finance Journal.
> Second, to further enhance label quality, we implemented a rigorous cross-review mechanism. In this process, three of our team members independently labeled the same dataset based on established standards. We then compared and analyzed the labeling results from each member, identifying and correcting potential differences and biases.
> #R4.A4:
> Thank you for your valuable suggestion. In our current work, due to computational limitations, we have not been able to fine-tune more of the newer open-source large models. As detailed in the appendix, a single A100 GPU currently cannot fine-tune
> Due to computational limits (a single A100 GPU), we couldn't fine-tune newer large models. However, our dataset consistently achieves high accuracy, even on smaller models, proving its value. Our focus is to build a comprehensive ESG dataset for future research. We will expand and fine-tune it on more models as our research progresses, inspiring further academic exploration. This dataset will help large models address real-world challenges like sustainability and social equity.
> #R4.A5:
> We will include a detailed analysis in the revised manuscript

---

### Official Review · Reviewer_8D8P · 2024-07-28

**Rating:** 5
**Confidence:** 4
**Clarity:** Yes.

**Review:**

### Pros:

1. The proposed benchmark lies on the domain that is important to the society yet underexplored by NLP researchers, thus I believe the proposed dataset is a nice addon to this direction.
2. Data is curated in a reasonable manner, particularly a standard ESG tree is used to annotate topics.

### Cons:

1. The title says “evaluating completeness of ESG reports”, while I feel there is a gap between predicting topic and quality labels of sentences and predicting completeness. Evaluating completeness of ESG reports is a complicated task which may still need expert assessment and the predicted labels from this paper may only serve as auxiliary inputs to assist in humans. I think it is necessary to clarify this gap and their relationship, otherwise it is hard to understand the significance of the proposed tasks.
2. The dataset is fully in Chinese if I understand correctly, then I think at least some Chinese pretrained LMs should be evaluated in the experiments – currently all the models are English-oriented that are known to be bad at Chinese.
3. Some experimental details are not described clearly, for example, Table 2 presents numbers without fine-tuning, is it from prompting? How are the numbers obtained? Because these are classification tasks, a common way is to compute logits of potential candidates and select the highest one, not sure how the authors conduct the experiments though.

**Strengths:**

See above.

**Additional Feedback:**

NA

**Correctness:**

For experiments design, I think the authors should at least include some Chinese pretrained LLMs given that the dataset is in Chinese.

**Documentation:**

I didn't find a URL to access to and judge the dataset.

**Limitations:**

This paper contains a limitation paragraph, yet it does not fully discuss the limitations, particularly the relation between predicting topics/quality and evaluating completeness of reports.

**Opportunities For Improvement:**

See the "Cons" above which may be addressed for improvement.

**Relation To Prior Work:**

Yes.

**Summary And Contributions:**

This paper introduces a new dataset for Environmental, Social, and Governance (ESG) reports. The authors claim that judging completeness of ESG reports is very important as many companies selectively disclose information in their reports, thus the authors create the proposed benchmark towards automatic completeness prediction. Specifically, the benchmark includes predicting ESG topics and classifying “quantitative” or “qualitative” labels for each sentence of a set of Chinese ESG reports. All the labels are manually annotated. Several pretrained models are evaluated on the proposed benchmark.

---

> ### Author Rebuttal · Authors · 2024-08-17
>
> Thank you for raising the issue. Response to # 8D8P
> #R3.A1:
> We evaluate ESG reports based on the integrity requirements proposed by the International Integrated Reporting Council (IIRC) (Breijer and Orij 2022) and the European Union's Corporate Sustainability Reporting Directive (CSRD) (Veltri 2020), incorporating existing research. Our evaluation considers both the thematic content and the quality of the text. Of course, this is only our attempt. We have invested significant human and time resources in compiling this dataset and sharing it with the academic community. This dataset serves as a valuable resource for researchers studying the integrity of ESG reports, helping to inspire greater interest and exploration in this field. Ultimately, it contributes to the development of creative solutions that enhance the integrity of ESG reports, thereby supporting corporate sustainability and improving societal and environmental outcomes.
> #R3.A2:
> Thank you for your observation. Our dataset was fine-tuned using Chinese pretrained models from HFL, optimized for Chinese text.
> #R3.A3:
> For these models, no prompts were used. We performed classification tasks by selecting the category with the highest probability as the final result. These details will be clarified in the revised manuscript to enhance transparency and understanding.

---

> > ### Comment · Reviewer_8D8P · 2024-08-29
> >
> > Thanks for your reply.

---

### Official Review · Reviewer_StBq · 2024-07-29
**wrong venue**

**Rating:** 3
**Confidence:** 4
**Correctness:** N/A
**Clarity:** yes

**Review:**

Significance: This topic seems quite niche for NeurIPS. The significance seems limited to assessing a particular type of document as part of financial assessments of companies. It does not have much interest beyond that one use case. Since this is not even a minor topic of interest among NeurIPS regulars, I think this is clearly the wrong venue for this paper. Furthermore, I found the justification for using LLM methods to perform this task thin and unconvincing. That evaluating these reports is difficult and requires domain experts does not necessitate the construction of datasets for training models. That sounds like the sort of job where human experts are the best resource for the job. The accuracy at assessing completeness is furthermore not that impressive, and surely that is only one of many things one would want to assess in such reports, so the tool doesn't seem particularly useful. Getting LLMs to read reports that are probably also written by LLMs more fluently is not an interesting or significant application of AI.

It's a clear reject for me based on significance, so I did not assess the other attributes.

**Strengths:**

The paper might be a fine contribution to a conference on a more niche topic.

**Additional Feedback:**

no further comments.

**Documentation:**

N/A

**Ethics:**

no concerns

**Limitations:**

The potential societal impacts of automating the evaluation of these reports are not even mentioned in the paper as far as I could tell, but I find the prospect of automating ESG report checking frankly depressing.

**Opportunities For Improvement:**

If there were something interesting for LLM work outside of this use case that came out of this research, I suppose it could begin to approach significance for NeurIPS.

**Relation To Prior Work:**

yes

**Summary And Contributions:**

The paper presents a benchmark for automatically assessing the completeness of Environmental, Social and Governance reports.

---

> ### Author Rebuttal · Authors · 2024-08-17
>
> Our topic is closely related to sustainable development and social and environmental protection, areas that are also the focus of many papers in the NeurIPS proceedings. Many studies (Morio and Manning 2023; Kaltenborn et al. 2023; Nguyen et al. 2023) have provided us with significant inspiration. For example, Morio and Manning (2023) constructed an NLP dataset on climate change, aiming to conduct evidence-based assessments of corporate climate policy engagement. Kaltenborn et al. (2023) built climate model datasets to predict new climate change scenarios.
> Furthermore, while the exceptional capabilities of large language models (LLMs) in text processing are well-established, their practical application in real-world, especially commercial scenarios, remains limited. Our team is committed to breaking this barrier by actively promoting the widespread application of LLMs in the field of environmental, social, and governance (ESG) assessment. This effort not only expands the application scope of LLMs but also introduces more intelligent and efficient solutions to ESG assessment, showcasing the promising integration of technology and sustainable development.

---

### Official Review · Reviewer_9fC5 · 2024-08-04
**ESG-related NLP Benchmark Dataset**

**Rating:** 4
**Confidence:** 4

**Review:**

The work introduces a new dataset for ESG completeness assessment. There is adequate level of originality. However, there are flaws in the methodology and writings of the paper.
- Regarding the ESG tree and ESG topics introduced, it would be beneficial for ablation studies on using different representations, compared to the final version used. Moreover, analysis of correlation and overlap between topics would also be useful to convince on the fitness of the proposed tree.
The data collected are from Chinese-listed companies and are in Chinese. This needs to be highlighted clearly. Moreover, it also limits the significance of the dataset to a wider community.
- An example of an ESG report and labelled data would be useful for the reader to follow more easily.
- Line 124: what is the agreement level between the annotators?
- Line 131: the claim “out-of-distribution sample set” is incorrect. The data are still from the same distribution, just without labels.
- Regarding the quality label: why is it called “quality label”?
- Paragraph 157 “ESG completeness evaluation”: what would be the score of a complete ESG report, and why? Moreover, Why “assign scores of 2 to “Quantitative” sentences and 1 to “Qualitative” sentences”?
- In the experiment, the baseline models are not the original versions introduced in the cited sources, but a Chinese version. This information is only in the Appendix and might mislead the readers.
- How did the inference on models without additional fine-tuning perform? Is it through prompting, and if so, what is the prompt?
- In section 4.4, the design of the prompts are not mentioned and make it not possible to understand and interpret the results.
- The experiments and discussions should be designed more towards the quality of the dataset itself. The current experiment results do not prove the quality of the labelling process of the dataset.
- Last but not least, the dataset in its current form is limited in having potential impacts. It is important to focus on having a quality labelling process with meticulous labels to enhance the dataset's practicality and impact in the field.

**Strengths:**

- Clear introduction to the problem.
- A detail Label Tree to provide fine-grained labels for the dataset
- Experiments with various models and model sizes.

**Additional Feedback:**

Please see above.

**Clarity:**

The structure of the paper is easy to follow. However, specific sections (e.g., concluding paragraphs of sections 3, 4.2, 4.3, 4.4) lack necessary information to understand. Additionally, some keywords are misused.

**Correctness:**

Some claims are incorrect, such as in line 131, and conclusions in section 4.4 are not clear enough. The construction of the dataset is questionable, as the annotation process is not detailed enough, such as missing agreement levels between annotators.

**Documentation:**

There is sufficient detail.

**Limitations:**

The discussion on the Limitation paragraph is misleading. The authors should consider other metrics beyond the performance of a deep learning model trained and evaluated on the dataset for evaluating the quality of the dataset itself.

**Opportunities For Improvement:**

Please see above. Additional experiments design toward the quality of the labels obtained through the labelling process, and revising and enhancing the clarity of the text are recommended. The impact of ESG report completeness is limited; it is important for the authors to strengthen the labelling process to achieve wider impacts in the field.

**Relation To Prior Work:**

The discussion of previous works is reasonable.

**Summary And Contributions:**

The paper discusses a new collection of 14468 ESG reports from Chinese-listed companies with topic and quality labels per sentence and evaluated the performance of pre-trained language models such as BERT, RoBERTa, and LLaMA2. Three PhD students are involved in the annotation process, producing 36 topic labels and 2 quality labels. The benchmarking result is promising for automated tasks in ESG completeness checking.

---

> ### Author Rebuttal · Authors · 2024-08-17
>
> We thank you for your invaluable  comments.
> A1: In this study, the ESG tree and ESG topics are not two different representations.
> We use the ESG tree (on page five of the main text, Figure 2) to define the completeness of the topic classification. Figure 2 illustrates the four-layer ESG tree we constructed, a hierarchical framework that dissects corporate sustainability into Environmental, Social, and Governance dimensions, each further divided into related sub-topics. Its 36 leaf nodes correspond to the 36 categories in our sentence topic classification tasks. The ESG tree facilitates a meticulous and systematic analysis of ESG topics. The hierarchical division of the ESG tree is derived from the standards of multiple ESG rating organizations.Additionally, thank you for your reminder. We will emphasize in the main text that the dataset is in Chinese.
> Due to space constraints, we chose to build Chinese data sets for reasons described inoverall rebuttal answer 1.
> #A2:
> We have included a detailed illustration of the text preprocessing process for the ESG report in the Rebuttal Appendix to make it easier for readers to follow.
> #A3: Again, this issue is of concern to all reviewers and we explain it in detail in the overall rebuttal answer 2
> #A4: Thank you for your reminder. We have carefully reviewed the entire text and have accordingly revised "out-of-distribution sample set" to "unlabeled sample", to ensure consistency and professionalism in terminology. These revisions enhance the accuracy and readability of the paper, enabling readers to better understand the data processing and analysis methods involved. Additionally, we conducted a thorough proofreading of the paper to ensure that no similar issues were overlooked. Your valuable feedback has been instrumental in improving the quality of our paper.
> #A5: There is growing encouragement for disclosing quantitative information (Hörauf et al. 2022). Therefore, we examine the quality of ESG text as a crucial component in assessing the completeness of ESG reports.
> We categorize ESG text quality into two classes: (1) ``Quantitative Text'', which reflects quantitative information about the ESG aspects of the company, and (2) ``Qualitative Text'', which reflects qualitative information about the ESG aspects of the company.
> #A6: Relationship Between Predicting Sentence Topics and Quality Labels and Completeness Quantification See overall rebuttal answer 3
> #A7:The large language models we used are based on the models from the original paper and are pre-trained Chinese versions developed by HFL. We chose these Chinese pre-trained models because they are better suited to the Chinese language environment of our dataset. We have clarified this and updated the references to ensure that readers accurately understand the models we used.
> #R1.A8:For models without additional fine-tuning, we directly use the original model for the classification task without employing any prompt. The process is straightforward; the model outputs probabilities for each class, and we select the class with the highest probability as the final result. The prompting process is utilized during the model fine-tuning in our work.
> #R1.A9:
> Thank you very much for pointing this out. For How Prompts Were Designed, see overall rebuttal answer 3
> #R1.A10:
> We fully understand the necessity of demonstrating the quality of our dataset, particularly the integrity of the labeling process. To address your concerns, we have provided additional details regarding the guidelines and validation procedures used in the dataset annotation.
> The ESG classification process was meticulously conducted following a set of well-defined principles and classification guidelines, as outlined in the attached document. In our ESG report classification task, we established specific standards for each category within the topic. These guidelines were developed by a postdoctoral researcher in law, based on authoritative ESG classification standards. This ensured that all annotators adhered to consistent criteria, thereby minimizing the potential for subjective bias. Moreover, the labeling was conducted by a master’s student in statistics, a Ph.D. in economics, and a postdoctoral researcher in law, and the annotation process included two rounds of review to ensure consistency and accuracy. This rigorous process further ensured the reliability of the labels.
> To validate the quality of the labeling, we conducted an inter-annotator agreement analysis. Initially, annotators reviewed each other's labeled data, followed by a final verification of the entire dataset by all annotators. This thorough analysis, combined with the strict adherence to the classification guidelines, strongly supports the reliability of our dataset.
> We are grateful for your feedback and will incorporate additional discussion of the dataset's labeling process and quality control measures in our revised manuscript.
> #R1.A11:
> The dataset we constructed holds significant importance for several reasons: First, it fills the gap in the availability of Chinese ESG report datasets, facilitating related research for more scholars. Second, it contributes to advancing the development of laws and regulations for ESG report disclosure standards in China. Third, our work on data collection, topic classification, and task definition principles aids in identifying and curbing corporate selective disclosure practices.
> Additionally, we have annotated sentences for 36 sub-categories of ESG. Figure 2 illustrates the four-layer ESG tree we constructed, a hierarchical framework that dissects corporate sustainability into Environmental, Social, and Governance dimensions, each further divided into related sub-topics. For example, the secondary indicators corresponding to the Environment include: Climate change, Natural capital, and Sustainable development management. Furthermore, the leaf nodes under Climate change are Carbon emissions and Response to climate change.

---

### Author Rebuttal · Authors · 2024-08-17

We thank reviewers for your invaluable and insightful comments.
1. Importance of Constructing a Chinese ESG Dataset:
First, while there are many open-access English ESG report datasets (Lee and Kim 2023; Tseng et al. 2013), there is currently no equivalent dataset for Chinese ESG reports. These reports are essential for both academia and industry to understand the contributions and investments of Chinese enterprises in ESG and sustainable development.
Second, our dataset facilitates the development of laws and regulations governing ESG report disclosure standards in China. By encompassing both the content and quality of ESG reports, it enables researchers to analyze the ESG performance of Chinese listed companies, identify existing challenges, and highlight best practices.
Third, scholars have noted that, due to the lack of standardized disclosure requirements in China, selective disclosure remains a significant issue (Wang and Liu 2024; Long et al. 2024). Our work on data collection, topic classification, and task definition is closely related to predicting the completeness of ESG reports and plays a critical role in identifying selective disclosure behaviors by companies.
2. Scientifically and Fairly Constructed Dataset to Reduce Bias:
First, the overall labeling process and the formulation of standards in this study (see Appendix Table 1) are coordinated by two experts in law and finance who specialize in ESG, CSR, and green finance research, with related work published in reputable journals such as the Journal of Financial Econometrics, Management Decision.
Second, to further ensure label quality, we implemented a rigorous cross-validation mechanism. In this process, three team members independently labeled the same dataset according to the established standards, followed by a comparison and analysis of the labeling results to identify and correct any potential discrepancies and biases.
3. Relationship Between Predicting Sentence Topics and Quality Labels and Completeness Quantification:
The International Integrated Reporting Council (IIRC) first proposed completeness requirements for ESG reporting (Breijer and Orij 2022). These requirements enhance topic breadth by integrating information on environmental, social, and corporate governance issues and emphasize the disclosure of quantified ESG information to improve integrity. Additionally, the European Union's Corporate Sustainability Reporting Directive (CSRD) mandates that a complete ESG report include both quantitative and qualitative information (Veltri 2020). Consequently, the topic coverage and text quality has become a crucial standard for assessing the completeness of ESG reports in management (Kennedy Nyahunzvi 2013; Darnall et al. 2022) and accounting (Melloni et al. 2017; Eng et al. 2021). For example, Melloni et al. (2017) evaluated the completeness of ESG reports by analyzing their coverage of ESG topics according to Bloomberg's ESG disclosure standards.

4. Chinese Pretrained LLM and Details on Models Without Fine-Tuning:
   The LLMs we used are based on the models from the original paper and are pre-trained Chinese versions developed by HFL. We chose these Chinese pre-trained models because they are better suited to the Chinese language environment of our dataset. For models without fine-tuning, we directly use the original model for the classification task without employing any prompt. The process is straightforward; the model outputs probabilities for each class, and we select the class with the highest probability as the final result. The prompting process is utilized during the model fine-tuning in our work.
5. How Prompts Were Designed:
 For the two tasks, we employed three distinct types of prompts:  1. Basic Prompt: This approach offers the simplest form, where the model is directly given the task type. An example of this would be: "Classify the ESG report text by topic."  2. Role-Based Prompt: Here, we assign a specific role to the model along with a concise task description. Imagine instructing the model as follows: "As a logical ESG report analyst, classify the ESG text according to the third-level indicators." Research has shown that setting a role for the model can significantly enhance its reasoning capabilities and the precision of its responses, especially when the model is tasked with assuming an expert identity (Kong et al. 2023; Xu et al. 2023; Chen et al. 2023; Zhao et al. 2023).  3. Detailed Prompt: This method provides the model with both an elaborate task description and a defined role, ensuring comprehensive instructions. For instance: "The ESG report comprises 3 first-level indicators, 7 second-level indicators, and 36 third-level indicators ..."
6.More Experiments with LLMs:  Thank you for your valuable suggestion. Due to computational limits (a single A100 GPU), we couldn't fine-tune newer large models. However, our dataset consistently achieves high accuracy, even on smaller models, proving its value. Our focus is to build a comprehensive ESG dataset for future research. We will expand and fine-tune it on more models as our research progresses, inspiring further academic exploration. This dataset will help large models address real-world challenges like sustainability and social equity.
7.We have added a new figure in the Rebuttal Appendix showing the text preprocessing process for the ESG report and sample labeled texts. Additionally, we have corrected the descriptive issues mentioned by the reviewers.

---

### Decision · Program_Chairs · 2024-09-26

**Decision:**

Reject

**Comment:**

This paper presents a dataset for predicting the completeness of Environmental, Social, and Governance (ESG) reports. While the topic falls within the scope of the DB track (though one reviewer suggested it might be off-topic), the paper has several major issues that place it below the standard, based on the reviews. The primary concerns are: (1) the data curation process lacks sufficient descriptions and is not reproducible. For instance, there is no explanation of how the 36 topics were selected, what the annotation agreement was, or how inconsistent and missing cases were addressed; (2) it is unclear whether the styles of ESG reports are generalizable to other countries; and (3) there is a lack of downstream use cases to demonstrate the practical values of the dataset, especially given that this is a domain-specific topic of interest.